# Extracellular Vesicles: A Crucial Player in the Intestinal Microenvironment and Beyond

**DOI:** 10.3390/ijms25063478

**Published:** 2024-03-20

**Authors:** Shumeng Wang, Junyi Luo, Hailong Wang, Ting Chen, Jiajie Sun, Qianyun Xi, Yongliang Zhang

**Affiliations:** Guangdong Provincial Key Laboratory of Animal Nutrition Control, National Engineering Research Center for Breeding Swine Industry, College of Animal Science, South China Agricultural University, Guangzhou 510642, China; wangsm9809@163.com (S.W.); luojunyi@scau.edu.cn (J.L.); wanghailong03@stu.scau.edu.cn (H.W.); allinchen@scau.edu.cn (T.C.); jiajiesun@scau.edu.cn (J.S.); xqy0228@scau.edu.cn (Q.X.)

**Keywords:** extracellular vesicle, intestine, microbiome, gut–brain axis

## Abstract

The intestinal ecological environment plays a crucial role in nutrient absorption and overall well-being. In recent years, research has focused on the effects of extracellular vesicles (EVs) in both physiological and pathological conditions of the intestine. The intestine does not only consume EVs from exogenous foods, but also those from other endogenous tissues and cells, and even from the gut microbiota. The alteration of conditions in the intestine and the intestinal microbiota subsequently gives rise to changes in other organs and systems, including the central nervous system (CNS), namely the microbiome–gut–brain axis, which also exhibits a significant involvement of EVs. This review first gives an overview of the generation and isolation techniques of EVs, and then mainly focuses on elucidating the functions of EVs derived from various origins on the intestine and the intestinal microenvironment, as well as the impacts of an altered intestinal microenvironment on other physiological systems. Lastly, we discuss the role of microbial and cellular EVs in the microbiome–gut–brain axis. This review enhances the understanding of the specific roles of EVs in the gut microenvironment and the central nervous system, thereby promoting more effective treatment strategies for certain associated diseases.

## 1. Introduction

Extracellular vesicles (EVs) are nanoparticles with a lipid bilayer structure that participate in intercellular communication. They are recognized as an important component of molecular nutrition and have also been harnessed for various applications, including as biological drug carriers for targeted delivery [1], diagnostic tools for specific diseases, and biological agents for therapeutic objectives [2]. EVs from different sources contain unique biological information in association with their parent cells and exert different effects on the intestine. Numerous studies have demonstrated that EVs from breast milk [3], plants [4], stem cells [5], and even the gut microbiota [6] have a positive effect on the intestine under normal physiological conditions. Intestinal health is associated not only with its own condition but also with the microbiota colonized within. The available evidence reveals that the vast majority of gut microbiota are composed of bacterial species, and the EVs they secrete can act as a potent immunomodulator in the host [7].

The discovery of the gut–brain axis has perfected the concept of designating the enteric nervous system (ENS) as our “second brain” [8]. Enteric glial cells (EGCs), the predominant cells in the ENS, are widely distributed throughout the intestine, and recent studies show that these cells can recruit immune cells and promote the self-renewal of intestinal stem cells to strengthen the intestinal barrier in response to a specific microenvironmental stimulation [9]. The interaction between the central nervous system (CNS) and the ENS has led to speculation regarding an association between the intestine and CNS diseases. Additionally, the interactions of EGCs with the gut microbiota have drawn attention to the potential influence of the gut microbiota on the gut–brain axis [10]. Consequently, this has led to the emergence of a new concept, the microbiome–gut–brain axis. Based on the evidence that EVs can cross the blood–brain barrier, the ubiquity of EV production across various cells including microorganisms, and the discovery of the microbiome–gut–brain axis, it is hypothesized that EVs may exert a certain impact on the microbiome–gut–brain axis.

Hence, this review provides insights into the basic facts and techniques of EV research and discusses how EVs of different origins are involved in the regulations of the intestine, the intestinal microenvironment, and other biological systems. We also explore how EVs participate in the microbiome–gut–brain axis and assess their potential as a novel therapeutic tool for CNS-related disorders.

## 2. Basics of Extracellular Vesicles

Extracellular vesicles (EVs) are nanovesicles with a lipid bilayer membrane-enclosed structure, secreted by various cell types. They generally comprise significant bioactive components, such as microRNAs, mRNAs, lipids, proteins, metabolites, and other signaling molecules that mediate intercellular communication pathways. In previous studies, extracellular vesicles have been classified into three categories based on their distinct production modes and sizes: microvesicles, exosomes, and apoptotic bodies [11]. Exosomes, ranging in size from 40 nm to 160 nm, are small EVs formed by endocytosis. Microvesicles, with a diameter of 150–1000 nm, are larger extracellular vesicles that separate from the cell membrane directly outward. Apoptotic bodies, around 50–1000 nm in size, are formed after programmed apoptosis, wherein the cell membrane bubbles to create apoptotic bulges that subsequently disintegrate into apoptotic bodies [12]. In 2018, the International Society for Extracellular Vesicles (ISEV) proposed the adoption of “EVs” to replace terms such as exosomes and microvesicles and apply operational prefixes to describe different subtypes of EVs, such as small EVs (sEVs), referring to their physical characteristics, CD63+/CD81+-EVs, named for their surface antigens, or hypoxic EVs, described for their conditions or cells of origin [13].

Exosomes, or sEVs, represent a prominent subset of EVs that have attracted considerable attention in scientific research [14]. They come from multivesicular bodies (MVBs), which are formed by the endocytosis of the plasma membrane. Subsequently, MVBs fuse with the cell membrane and release the intraluminal vesicles (ILVs), which are formed by the inward budding of the late endosomal membrane existing in the MVBs, to the outside of the cell. These ILVs in the extracellular space are defined as exosomes [13] (Figure 1). Previous studies believed that the biogenesis of exosomes was related to tetraspanins and the endosomal sorting complexes required for transport (ESCRT). CD81, CD82, and CD9 are the three tetraspanins that play a critical role in the sorting of various cargoes into exosomes [15], while CD9 and CD63 are considered to be related to the formation of exosomes. An ESCRT consists of four subunits (ESCRT 0, ESCRT I, ESCRT II, ESCRT III) and accessory proteins (TSG101, ALIX, Vps4, and VTA1). ESCRT subunits are absorbed in the later stage of exosome formation, but accessory proteins such as TSG101 and Alix will remain [16]. These tetraspanins and accessory proteins are now considered marker proteins of exosomes. The ESCRT-independent formation mechanism of exosomes has also been discovered. Research has revealed that EVs derived from mouse oligodendroglial cells transfer exosome-associated domains depending on sphingolipid ceramide rather than ESCRT. Inhibition of neutral sphingomyelinase has been found to decrease the release of EVs [17]. Subsequent studies have also demonstrated that sphingolipid metabolites are involved in the formation and sorting of cargoes of ILVs in EVs [18]. Figure 1 summarized the biogenesis and the compositions of EVs. 

## 3. EVs in Health and Disease

Given that EVs can be transported via the circulatory system to various parts of the body, carrying bioactive substances and nucleic acid molecules that facilitate communication between cells, EVs serve a vital function in both maintaining health and contributing to disease processes. In a healthy state, EVs play a vital role in maintaining the stability of the internal environment and normal tissue function by promoting communication and material exchange between cells. For example, mesenchymal stem cell EVs can treat tissue fibrosis and promote tissue regeneration by delivering signaling molecules [19]. EVs can modulate the immune system, regulating the activity and function of immune cells by carrying immune activation or inhibition signals [20]. The miRNAs of circulating EVs from adipose tissue can regulate systemic metabolism and mRNA translation in other tissues, indicating that EVs can also maintain metabolic balance in the body [21]. EVs have been shown to participate in the communication between glial cells and neuronal cells, and it is believed that EVs may change the morphology and function of target cells when transmitting information, regulating homeostasis, immune function and inter-glia communication. It plays a key role in the transfer of biomolecules [22]. However, as a carrier that can target systemic organs, the functions of EVs are complex and diverse, and there are also potential harmful effects. Studies have shown that miRNAs in senescent cell EVs can inhibit macrophage responses, prevent neutrophil recruitment, and promote inflammation. In SEVs, The protein may lead to innate immune senescence and inflammation [23]. Plasma EVs in patients with severe dengue fever carry pro-inflammatory and anti-inflammatory factors and significantly increase the levels of IFNγ, TNF-α, and IL13. Plasma EVs can promote immune cells in peripheral blood by stimulating peripheral blood. inflammatory response to aggravate the inflammatory response [24]. After traumatic brain injury, brain-derived EVs transport pathogens and cause secondary damage, and the inflammasome-related proteins carried by EVs can cross the blood-brain barrier and reach remote tissues to cause inflammatory dysfunction [25]. Previous studies have shown that EVs are related to the aggregation and secretion of susceptibility proteins in neurodegenerative diseases, such as β-amyloid, α-synuclein and prion protein in Alzheimer’s disease [26]. EVs from highly metastatic melanoma lead to primary tumor metastasis through the receptor tyrosine MET [27]. In addition to the potential harmful effects of EVs under pathological conditions, EVs can also be used as biomarkers for the diagnosis and prognosis of early diseases because the molecular characteristics they carry reflect the pathophysiological state of the source cells. EVs accumulate within atherosclerotic plaques, so it is speculated that the concentration of specific plasma EVs can be used to judge the severity of the plaque [28]. As early as 2015, studies showed that glypican-1 is specifically enriched in EVs derived from cancer cells, so GPC1 circulating EVs can be used as a potential non-invasive diagnostic and screening tool for early pancreatic cancer [29]. Whether they are regulating the immune system, promoting tissue repair, or carrying bioactive substances to affect target cells in target organs, these show that EVs have a wide range of biological functions in the body, and the impact of EVs is not limited to adjacent cells, but also affects target cells. Including interactions between different tissues and organs in the body, such as the intestines. As one of the most important organs of the human body, the health of the intestine directly affects the health of the entire body. Therefore, the impact of EVs on the intestine deserves special attention, as they play an important role in intestinal health, disease, and intestinal microbial communities.

The intestine, being the organ with the most extensive surface area in the body, is considered to be vital in the digestion of food and the provision of nutrients to sustain proper functioning. The mucous barrier on the surface of the intestine serves as an initial defense in intestinal protection [30]. Numerous microorganisms are colonized in the intestine, and sustain a dynamic equilibrium of their own composition and function via a negative feedback mechanism [31]. The gut microbiota interact symbiotically with the host, and the gut microbiota can produce diverse metabolites from dietary sources or host endogenous compounds. Intestinal microbes can encode glycosaminoglycan degradation genes, ferment complex polysaccharides to produce short chain fatty acids (SCFAs), synthesize specific lipopolysaccharides (LPS) and biosynthesize some essential amino acids and vitamins [32], which are closely related to host metabolic disorders. Intestinal microbes are also in-volved in the metabolism of bile acids and affect the host’s lipid metabolism and glucose metabolism [33]. Therefore, intestinal microbes are called central regulators of host metabolism [34]. They are crucial in the immune system by promoting the differentiation of the host cells to defend against external pathogens. Gastrointestinal microbiota can affect T cell differentiation, for example, butyric acid in SCFAs can promote peripheral induced regulatory T cell differentiation, thereby inhibiting systemic inflammation [35]. In addition, the gut microbiota also aid the digestion of food in the gastrointestinal tract, enabling the breakdown of nutrients that would otherwise be unaccessable to the host [36]. 

As an absorptive organ, the intestine not only receives endogenous EVs from other tissues, cells, blood and intestinal microorganisms circulated in the body, but also receives exogenous EVs from ingested foods. EVs, produced by almost all cells, show varied functions that much resemble those of their parent cells. For example, camel milk-derived EVs have demonstrated anticancer properties and immuno-reactivities similar to camel milk, but even better in the former and less in the latter [37]. Under normal physiological conditions, EVs on the apical and basolateral surfaces of the intestinal epithelium play a crucial role in immune regulation. EVs in intestinal epithelial cells may contain cofactors involved in antigen presentation, and the presence of EVs allows intraluminal antigens to be presented to the intestinal lamina propria and even to T cells throughout the body [38]. This mechanism effectively links antigen recognition in the intestinal lumen with the systemic immune system, promoting coordinated immune responses between the intestine and antigens. However, in patients with inflammatory bowel disease (IBD), EVs in the intestinal lumen carry pro-inflammatory factors at a greater abundance than that of the healthy control group, resulting in immune dysfunction [39]. Hence, the regulatory effects of EVs on the intestinal system are diverse and complicated as it could receive EVs of multiple origins. In recent years, research has covered the regulatory effects of endogenous EVs such as mesenchymal stem cell-derived EVs (MSC-EVs), and exogenous EVs such as milk EVs (Figure 2).

Mesenchymal stem cells (MSCs) are multipotent stromal cells that can differentiate into a variety of cell types and are found in a variety of tissues, including bone marrow, fat, umbilical cord blood, and dental pulp. MSCs regulate through secreting paracrine factors. Amongst these factors, MSC-EVs have emerged to be a crucial component of intercellular communication [40]. These EVs have similar biological functions to MSCs, while exhibiting better biostability, targeting capability, biocompatibility, and low immunogenicity and toxicity. Additionally, MSC-EVs are not susceptible to cellular aging or potential tumorigenicity, nor do they raise moral and ethical concerns [41]. Increasing studies have shown that MSC-EVs have robust anti-inflammatory and immunomodulatory properties, as well as distinct capabilities in tissue repair, thereby significantly contributing to the maintenance of intestinal health. For example, human umbilical cord MSC-EVs have been found to effectively mitigate IBD by repairing the intestinal mucosal barrier and maintaining immune homeostasis via tumor necrosis factor-α-stimulated gene 6 (*TSG-6*) [42]. MSC-EVs containing miR-181a is capable of increasing the expression of intestinal barrier-associated genes, Zonula Occludins-1 (*ZO-1*) and Claudin-1, while decreasing the levels of inflammatory factors Tumor Necrosis Factor-alpha (TNF-α), Interleukin 6 (IL-6), and Interleukin 17 (IL-17). This enhanced anti-inflammatory capability also affects the gut microbiota, further postulating that the protective effect of MSC-EVs in the colitis mouse model is associated to their regulation of the gut microbiota [43]. Some studies have demonstrated that the impact of certain MSC-EVs on the intestine is mediated through macrophages. For instance, MSC-derived EVs polarize colonic macrophages to M2b phenotype to attenuate inflammatory response and preserve intestinal barrier integrity [44]. It has been demonstrated that oral administration of layer-by-layer encapsulated MSC-EVs promotes EV absorption by macrophages and intestinal epithelial cells, enabling EVs to exert anti-inflammatory and reparative effects, ultimately alleviating colitis in mice [45]. Furthermore, umbilical cord MSC-EVs alleviate colitis in mice by slowing down macrophage pyroptosis by inhibiting cysteine caspase11/4 [46]. 

Macrophages play a critical role in intestinal homeostasis and are key regulators of the intestinal immune system. Their main role involves distinguishing between potential pathogens and harmless antigens present in the intestinal environment, avoiding immune responses to benign substances such as food particles and commensal bacteria, while ensuring a rapid and effective response to harmful pathogens. Macrophage within the intestine is necessary for the regeneration of the intestinal epithelium and the preservation of immune homeostasis in the intestinal mucosa. The polarization of macrophages into either M1 pro-inflammatory or M2 anti-inflammatory phenotypes depends on the environmental conditions in which they reside. Macrophage-derived EVs can interact with intestinal epithelial cells as well as other cells, and studies have shown that EVs derived from M2 anti-inflammatory macrophages can enhance intestinal mucosal healing and attenuate colitis [47].

Plants also produce nanoparticles similar to extracellular vesicles with comparable dimensions and shapes, and they encapsulate proteins, lipids, nucleic acids, and metabolites. However, these nanoparticles are not spontaneously released by plants and are not entirely identical to animal EVs in structure. Plant exosome-like nanoparticles (ELNs) differ in their structure of lipid membranes compared to mammalian EVs, with the former being rich in phosphatidic acid and phosphatidylethanolamine, whereas the latter exhibiting higher levels in cholesterol. Some have reported that plant ELNs contain antibacterial compounds, such as proteins, sRNAs, and lipid signals that aid in resisting pathogen invasion, and form defense structures in close proximity to the infected site [48]. Plant-derived ELNs have also been the subject of scientific investigations in treating intestinal-related diseases, as the intestine are easily exposed to ELNs during the consumption of edible plants [49]. However, the relationship between food-derived vesicles such as plant exosome-like nanoparticles ELNs and the intestinal barrier and immune system is complex and multifaceted. These ELNs can indeed carry antigens, including proteins, which can have profound effects on receptor cells in the gut. Experiments have shown that grapefruit can load exogenous bovine serum albumin (BSA) and heat shock protein 70 (HSP70) and deliver them to human cells. This study shows that plant-derived EVs can indeed cross species barriers and deliver functional proteins to human cells, including those lining the gut [50]. Previous studies have demonstrated that various edible plant-derived ELNs are involved in signaling pathways in interspecies communication and intestinal homeostasis maintenance. For example, ginger derived ELNs have been found to activate nuclear factor (erythroid-derived 2)-like 2 (Nrf2) and induce the production of anti-inflammatory molecules, thereby alleviating intestinal inflammation [51]. ELNs of edible mulberry bark have been shown to reduce cellular damage associated with colitis by promoting the activation of the heat shock protein family A member 8 (HSPA8)-mediated anti-microbial peptides (AhR) signaling pathway [52]. The lipid components of grape ELNs induce Lgr5+ stem cells to protect mice against colitis [53]. Garlic EVs have been found to upregulate the expression of barrier-related proteins and inhibit the excessive production of pro-inflammatory cytokines in Caco2 cells induced by LPS, suggesting their potential utility in IBD therapy [54]. Although it is surprising that ELNs from plants can be delivered to cells and affect the host, complex interactions within the intestinal ecosystem explain this possibility. The intestinal barrier not only functions as a physical barrier, but also actively binds external EVs, including EVs from dietary sources such as plants, to regulate immune responses and maintain a balance between accepting benign substances and resisting pathogens. This complex regulatory network involving endogenous and exogenous EVs highlights the potential of these vesicles to influence human intestinal health and disease.

Milk EVs have recently attracted considerable attention due to their potential therapeutic effects and reduced immunogenic properties compared to artificial nanoparticles [55]. Breast milk serves as an essential source of nutrients and immune components for the development and well-being of the intestine, especially for newborns. The precise mechanism by which milk EVs remain intact through the gastrointestinal tract, despite the exposure to acidity and digestive enzymes, is not apparent. This may be partially due to the existence of calcium in the lipid membrane, which is known to contribute to membrane stability [56]. Once entering the intestine, milk EVs rely on their surface glycoproteins for entry into intestinal cells, endothelial cells, and immune cells via endocytosis, ultimately accumulating in peripheral tissues [57]. Given that milk derived EVs are predominantly released through budding of the membrane of breast epithelial cells, it is speculated that the biological molecular information within, such as miRNA, originates from breast epithelial cells [58]. Recent studies have shown that there is no significant difference in the proteins and mRNA contained in EVs in the breast milk of premature and full-term infants, but the immunomodulatory substances carried by EVs are significantly different. It is speculated that premature milk EVs are secreted by immune cells, while those of full-term infants of breast milk EVs are more likely to come from mammary epithelial cells [59]. However, it is widely believed that breast milk harbors a diverse and abundant microbial community originating from the surface skin, infant oral contamination, and the gut-breast axis, inferring that milk EVs could also contain a certain number of microbiota-derived EVs, which are involved in neonatal intestinal immune formation and colonization of gut microbiota, and serve as receptors for bioactive molecules in host cells [60]. Nonetheless, the impact of milk EVs on the gut is still of significant importance. Milk EVs have been shown to improve intestinal permeability and intestinal structure, and promote cell proliferation. In addition, miRNAs in milk EVs are critical in intestinal maturation, barrier function, and inhibition of nuclear factor-kappa B (NF-kB) signaling [61]. Milk EVs and their miRNAs can enter the circulatory system to influence the epigenetics of multiple organs and modulate a variety of nuclear receptors by controlling cellular regulatory factors [58].

Interestingly, EVs in milk vary across different species and different periods. Yak milk EVs surpass regular cow milk EVs in abundance by 3.7 times. They also exhibit a higher capability to activate the hypoxia-inducible factor signaling pathway, reduce the occurrence and severity of intestinal inflammation, attenuate hypoxic injury, and promote the survival of intestinal epithelial cells [62,63,64]. Research on EVs in milk from different periods shows that both preterm and term infants could absorb milk EVs via intestinal epithelial cells. However, breast milk EVs from mothers of preterm infants are of higher abundance and exhibit a more pronounced promotion in the proliferation of intestinal epithelial cells [65]. Compared with mature milk, colostrum-derived EVs are significantly higher in lactoferrin and casein, which results in a more potent anticancer effect by inducing apoptosis and inhibiting inflammation [66]. The breadth of miRNAs in colostrum also suggests a greater diversity of miRNAs in colostrum EVs than in mature milk EVs, though there are research pointing out that it is difficult to completely distinguish colostrum and mature milk during sample collection [67].

## 4. Microbiota Involved in and beyond the Intestine

Hundreds of millions of microbiota reside in the intestine, which encode millions of genes, far exceeding the number of genes in the human genome. They provide unencoded enzymes to participate in the host’s metabolism and engage in interactions with the host that contribute to the maintenance of immune equilibrium. To ensure intestinal homeostasis, the gut microbiota conduct a benign competition, enabling the host to sustain a stable, appropriate balance of energy and lipid metabolism and an effective intestinal barrier [68]. Bacteria in close proximity to the host can influence the host through direct contact, whereas those located at a greater distance rely on secreting soluble biochemical substances to interact with the host, such as peptides, proteins, nucleic acids, lipopolysaccharides, membrane vesicles, and so on [69]. As the most prominent member of the intestinal microecology, the intestinal microbiota is deeply involved in the intestinal health of the host [70] (Summarized in Table 1). Altering the composition of the gut microbiota can increase the permeability of the intestinal barrier and trigger an inflammatory response [71]. EVs from various sources, including EVs secreted by the microbiota itself, have been shown to have profound effects on the intestinal barrier, immune response, and overall intestinal health, and this intricate relationship is bidirectional. For example, it has been shown that unhealthy dietary patterns may promote gut microbiota dysbiosis and increase the production of harmful gut microbiota-derived EVs. These EVs may impair intestinal barrier permeability and intestinal inflammation, leading to the progression of metabolic diseases such as obesity and diabetes. Conversely, enhancing beneficial gut microbiota extracellular vesicles through a healthy dietary pattern or taking probiotics could provide therapeutic strategies to counteract these effects [72]. For example, the common fungal toxin, deoxynivalenol (DON), in food can cause damage to the host’s intestinal microbiota, polarization of M1 macrophages, and damage to the intestinal barrier. Probiotic bacteria, *Lactobacillus murinus* (*L. murinus*), can convert M1 polarization to M2 polarization, reduce pro-inflammatory factors, and improve anti-inflammatory factor levels. In addition, its secreted EVs enhance M2 macrophage polarization by activating TLR2, reversing the slow growth and barrier damage caused by DON [73]. The mechanisms that promote the translocation of microbiota-derived EVs in intestinal epithelial cells have also been elucidated, including endocytosis by IECs, transcytosis of the lamina propria, and interactions with resident immune cells. This process highlights the ability of EVs to cross biological barriers in the gut and directly influence immune responses [74].

The intestine is constantly receiving EVs from different sources and with different functions, so these EVs may also have different effects on the gut microbiota (Figure 3). In 2019, Zhou first reported the potential of bovine milk EVs in modulating microbial communities in mice [75], and subsequent studies further revealed that milk EVs can increase the abundance of colonic microbiota in mice, particularly *Bifidobacterium*, *Dubosiella*, and *Lachnoclostridium* [76]. Studies have demonstrated that milk EVs, both human- and bovine-derived, can influence intestinal immune responses and intestinal microbial communities. The positive effects of these alterations are not fully understood, but a vast majority of studies have shown that milk EVs can increase the abundance of beneficial bacteria while diminishing harmful bacteria in mouse models. Although the extent of these changes varies in male and female mice, a general pattern remains consistent, that these changes in microbiota will affect the intestinal level of metabolites produced by the microbiota, especially SCFAs [77], and increase intestinal epithelial absorptive cells [78]. SCFAs produced by the gut microbiota are mainly propionate, acetate, and butyrate, which not only regulate the growth of the intestinal microbial community but also provide energy for the gut microbiota and the host’s intestinal barrier cells [79]. In colonocytes, butyrate serves as a primary energy source and increases the synthesis of tight junction proteins. Acetate and propionate are substrates for adipogenesis and gluconeogenesis in peripheral tissues [80].

Other food-derived EVs can also directly affect the gut bacteria due to their direct contact with the intestinal lumen. Lemon-derived ELNs have been found to protect the intestinal microbiota from bile damage and increase the abundance of the *Lactobacillus rhamnosus* GG [81]. Similarly, ELNs from Camellia, an edible tea flower, can increase the abundance and diversity of the gastrointestinal microbiota [82]. Ginger ELNs have been shown to mitigate colitis by promoting the colonization of *Lactobacillus rhamnosus* and producing ligands for the aryl hydrocarbon receptor via endogenous miRNAs [83]. Interestingly, gut microbiota across different species have different absorptive preferences for the ELNs of various plant species. For instance, ginger ELNs are preferentially absorbed by *Lactobacillaceae*, while garlic and grapefruit ELNs are preferentially absorbed by ruminants [84].

As previously mentioned, EVs derived from MSCs from the human umbilical cord or placenta have been demonstrated to inhibit inflammation. Consequently, the attenuation of the inflammatory response improves the structure of the colonic wall in mice, reduces the abundance of pro-inflammatory bacteria such as *Akkermansia muciniphila* and *Escherichia coli*, and ultimately restores the homeostasis of gut microbiota [40]. Recent studies have shown that MSC-derived EVs can improve the composition of the gut microbiota by significantly elevating OTU abundance and mitigating colitis-induced decrease in α-diversity while increasing the percentage of beneficial intestinal bacteria and counteracting the deleterious function of disease-associated bacteria [85]. The administration of EVs secreted by embryonic MSCs can reduce the presence of lipopolysaccharide [86] in the intestine and increase gut microbiota SCFAs, a mechanism which effectively prevents the impairment of intestinal barrier function caused by LPS and indirectly diminishes the recruitment of adipose tissue macrophages (ATM) triggered by LPS [87]. Furthermore, SCFAs have been shown to be associated with ulcerative colitis attenuation [88]. Interestingly, not all EVs have positive effects on the microbiota. EVs released from the intestinal mucosa of IBD patients contain defense proteins from the host, and upon uptake of these EVs by the intestinal microorganisms, an adaptive reaction occur, resulting in a dynamic imbalance of gut microbiota, and finally trigger mucosal inflammation of greater extent [89].

The consequences of gut microbiota alterations are not limited to the intestinal system only, but can also affect other downstream cells, organs, and systems. *Parvimonas micra*, which exists in oral mucosa and the gastrointestinal tract, promotes the progression of colon cancer by upregulating miR-218-5p in cells and EVs, inhibiting protein tyrosine phosphatase receptor R (PTPRR), and subsequently activating the Ras/ERK/c-Fos signaling pathway upon cocultivation with human colon cancer cells [90]. Adherent-invasive *Escherichia coli* (AIEC) can inhibit let-7b in EVs of intestinal epithelial cells to promote the fibrotic phenotype of intestinal macrophages, and then promote the formation of fibrosis [91]. In addition, the disruption of gut microbiota can induce the release of EVs by intestinal epithelial cells, causing M1 polarization of macrophages in the mesenteric lymph nodes in the immune system. M1 macrophages subsequently participate in the local immune response, resulting in increased circulating inflammatory cytokines and leading to sepsis [92]. The change in gut microbiota can also result in the destruction of the intestinal epithelial barrier, facilitating the entry of microbiota and its derived factors into the circulatory system. This could induce severe systemic inflammation, and possibly affect the blood–brain barrier [93]. Recently, it has been found that intermittent hypoxia exposure in mice results in an alteration in the composition and metabolism of the gut microbiota, which consequently increases intestinal permeability [94], affects the composition of plasma EVs, disrupts the balance of adipocytes, and ultimately leads to metabolic dysfunction [89].

## 5. Microbiome–Gut–Brain Axis

In the past, early research observed a bidirectional hormone and neural signaling pathway connecting the gastrointestinal tract and the central nervous system (CNS), which has been named the gut–brain axis. However, this description primarily focused on the conventional influences of nutrition-induced neural and hormone factors, neglecting the role of gut microbiota on the intestine [95]. Within the last 15 years, with the significant advances in microbiology, the comprehension of the gut–brain axis has gradually evolved into a systematic biological perspective of an interaction between the microbiota, gut, and brain [96]. The microbiome–gut–brain axis consists of six key components: the nervous system, the hypothalamic–pituitary–adrenal axis (HPA), the neuroendocrine network, the gut microbiota and its metabolites, the gut immune system, and the physiological barrier system. Within this axis, the gut microbiota is recognized as a relatively independent and diverse mediator, interacting with other components through neuroanatomical, neuroendocrine, intestinal endocrine, neuroimmune, and metabolic pathways [97]. The influence of gut microbiota on CNS function, including certain emotional responses and behaviors, has been reported. One study demonstrated that intestinal microbial metabolites, SCFAs, affect the central nervous system by regulating neural plasticity and gene expression. In addition, intestinal microorganisms can not only synthesize neurotransmitters themselves but also induce the host to produce neurotransmitters [98]. Studies have elucidated the role of metabolites such as SCFAs and branched-chain amino acids in mediating information between the gut microbiota and the brain via a variety of approaches, such as tryptophan metabolism, the vagus nerve, and the enteric nervous system (ENS) [99], and may contribute to neurological diseases such as anxiety, obesity, Parkinson’s disease, and Alzheimer’s disease [100].

As a promising drug transport carrier, EVs participate in various physiological and pathological processes. For example, certain EVs from specific sources have been proven to pass the BBB, which is a highly selective barrier to transverse, thereby improving the low transcytosis rate of the BBB by modulating the endocytic pathway of brain endothelial cells [101]. This transmission of information via EVs between the gut and brain contributes to various physiological development or pathological changes [102]. Notably, microorganisms can produce EVs. Hence, it is presumed that microbiota EVs contribute to the intricate interplay of the microbiome–gut–brain axis by facilitating the transfer of bioactive substances between cells, potentially causing adverse effects such as toxicity or immune stimulation, while also exerting positive effects on cell activity and biological function. They are critical in intestinal mucosal signaling processes and play a role in maintaining intestinal homeostasis [72].

The EVs of Gram-negative bacteria, initially discovered in 1960, are produced by outer membrane budding and are referred to as outer membrane vesicles (OMVs). Subsequent research revealed the existence of four main types of microbiota vesicles: OMVs, produced by outer membrane budding; outer inner membrane vesicles (OIMVs) and explosive outer membrane vesicles (EOMVs), produced by explosive cell lysis; and cytoplasmic membrane vesicles (CMVs), produced by Gram-positive bacteria [103]. Unfortunately, the precise mechanisms of the release of certain microbiota vesicles remain unclear [104]. Bacteria-derived EVs generally encompass lipids, nucleic acids, proteins, and other small molecules. Via microbial EVs, the biomolecules within the microbiota are transferred to other tissues, organs, or cells of the host to modulate the synthesis and secretion of proteins, nucleic acids, lipids, or other metabolites in the recipient cells. However, the impacts of different strains of microbiota on the host exhibit some degrees of uncertainty and diversity [105]. Some bacteria-derived EVs contain proteins capable of modulating the host immune system, but the content of these substances is contingent upon the classification and specific characteristics of the parent species [74]. EVs from intestinal probiotics and commensal E. coli activate dendritic cells (DCs) in the gut in a strain-specific manner, resulting in the activation of the toll-like receptor (TLR) signaling pathway, the regulation of miRNA expression, and the release of immune mediators in cellular EVs, which coordinate the stimulation of appropriate T cell responses [106].

The uptake of microbial EVs by intestinal epithelial cells occurs via various mechanisms, including actin-dependent macropinocytosis, clathrin-mediated endocytosis, caveolin-mediated endocytosis, or clathrin- and caveolin-independent mechanisms [74]. Some EVs also migrate directly across the intestinal epithelial barrier through the paracellular pathway to enter the circulation [107]. The pathway of microbial EV uptake and entry kinetic efficiency is determined by the composition of the bacterial cell wall. For example, the lipopolysaccharide O antigen on the outer membrane of Gram-negative bacteria can guide OMVs to undergo raft-mediated endocytosis, accelerating the uptake of intravesicular cargo and delivery [108]. Clathrin-mediated endocytosis is the primary pathway for Helicobacter pylori vesicle uptake [109]. There is speculation that microbial EVs participate in the microbiota–gut–brain axis in three distinct approaches: firstly, through vagal regulation. EVs of *Lactobacillus rhamnosus* can stimulate afferent nerves of the ENS [110]; secondly, through endocrine regulation. *Akkermansia muciniphila* and its EVs in the gut can enhance serotonin concentration and influence serotonin signaling through the gut–brain axis [111]; lastly, through the regulation of body fluid circulation, some bioactive components are encapsulated into EVs and delivered to the CNS.

EVs produced outside the microbiome–gut–brain axis also exert a particular impact on certain diseases associated with the axis. EVs secreted by murine bone marrow MSCs induce alterations in the gut microbiota composition, specifically *Proteobacteria*, *Muribaculaceae*, *Lachnospiraceae*, and *Acinetobacter*. These alterations in the gut microbiota play a beneficial regulatory role in cerebral hemorrhage [112]. On the other hand, the disturbance of gut microbiota in mice with Alzheimer’s disease inhibits the neuroprotective effect of MSC-EVs, while the administration of antibiotics amplifies the therapeutic benefits of MSC-EVs by altering gut microbiota composition, affecting tryptophan metabolism, enhancing intestinal barrier function, reducing intestinal permeability, and impeding bacterial infiltration from the intestinal lumen and diffusion throughout the body, hence collectively attenuating neuroinflammation in Alzheimer’s disease [113]. In addition, EVs modified by rabies virus glycoprotein (RVG-EVs) have shown an affinity to the brain, which has led to its utilization in the treatment of Parkinson’s disease. The latest research has demonstrated that RVG-EVs allow for repetitive targeting of the brain and avoid delivery to sites other than the brain. They also transmit information between the gut and the spinal cord, thereby down-regulating α-synuclein expression in the brain, intestine, and spinal cord, and ultimately alleviating Parkinson’s disease [114,115].

## 6. Conclusions and Perspectives

Recent advances have revolutionized our understanding of EVs from viewing them as mere cellular byproducts to recognizing their critical role in cellular communication. The ability of EVs to cross biological barriers and their precise delivery emphasizes their potential in drug delivery [116], neurodegenerative diseases, and targeted therapy [117,118]. These properties are further demonstrated in the intestine, where EVs affect not only intestinal cells but also the gut microbiota. This interaction plays a crucial role in the ontogeny of the gut and has been further complicated by the discovery of the microbiota-–gut–brain axis, suggesting that EVs may be key mediators in this complex network.

However, challenges remain in identifying EV subtypes and understanding their specific roles, requiring further investigation of their biogenesis and interaction mechanisms [119], which are critical to developing their therapeutic potential. The dynamic nature of the gut microbiota and the complexity of the microbiome–gut–brain axis further complicate the understanding of the precise role and impact of EVs. It is worth noting that, although it has been established that almost all cells can produce EVs, and many studies have focused on EVs produced by the CNS to explore the mechanisms of their effects in the body, as part of the microbiota–gut–brain axis, can the ENS produce EVs, and if it can produce EVs, what impact will it have on the microbiota–gut–brain axis? Although the study of enteric nervous system-derived EVs faces many difficulties, such as the complicated and cumbersome steps to extract primary enteric nerve cells, the difficulty in expanding culture due to the characteristics of the nervous system, and the difficulty in extracting EVs, enteric nervous system-derived EVs are still worth exploring and discovering, which is crucial for perfecting the concept of the microbiome–gut–brain axis. Therefore, future studies lie in elucidating the specific functions of EVs within the enteric nervous system and the mechanisms by which gut microbiota-derived EVs affect the central nervous system.

In summary, the specific functions of EVs in the microbiota–gut–brain axis must be further investigated to elucidate the potential role of intestinal EVs in CNS-related diseases. The possibility of using EVs to modulate the gut and intestinal microenvironment, or even influence other physiological systems such as the central nervous system, is exciting. Continued research in this area promises to unravel complex intercellular communication pathways and provide new insights into the management of health and disease in the nervous and gastrointestinal systems.

## Figures and Tables

**Figure 1 ijms-25-03478-f001:**
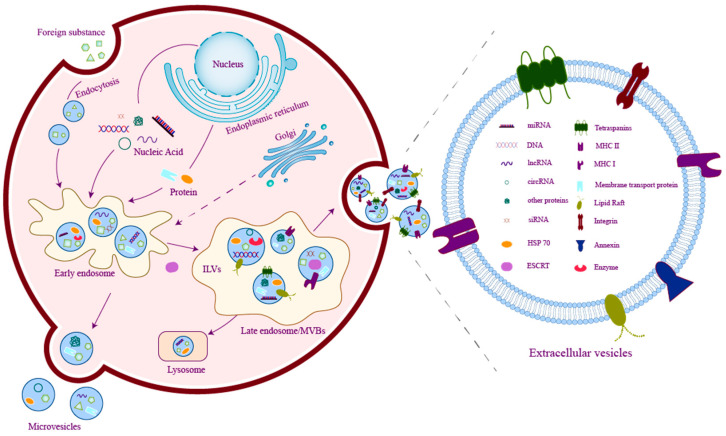
The generation and composition of EVs.

**Figure 2 ijms-25-03478-f002:**
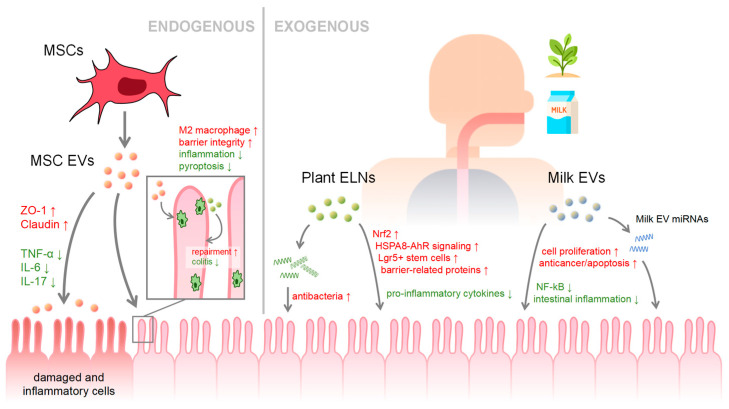
Mechanisms and effects of certain EVs on the intestine. Red and green arrows indicate an upregulation or downregulation of the specific gene/protein/pathway.

**Figure 3 ijms-25-03478-f003:**
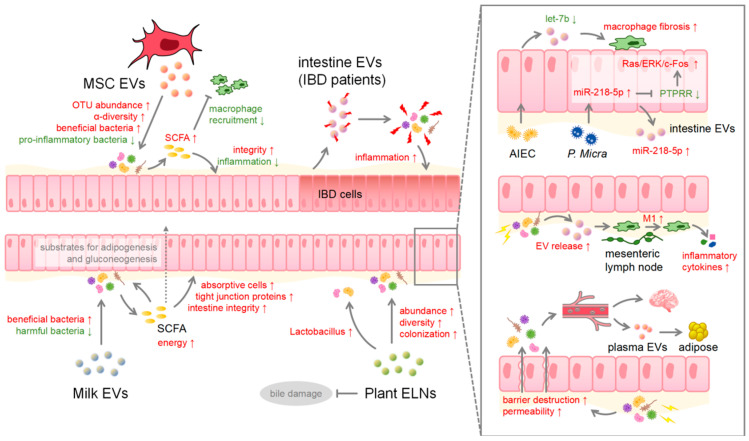
EV and microbiota interaction involved in intestinal and bodily health. Red and green arrows indicate an upregulation or downregulation of the specific gene/protein/pathway.

**Table 1 ijms-25-03478-t001:** Different types of microbiota and their effects in the intestine.

Microbiota Type	Role in the Intestine
*Bifidobacterium*	Decompose SCFAs produced by dietary fiber, improve the intestinal environment, help the absorption of nutrients, interact with intestinal immune cells, and regulate immune responses.
*Dubosiella*	Maintain the integrity of the intestinal mucosal barrier and regulate the diversity of the intestinal microbial community
*Lachnoclostridium*	Break down dietary fiber and other non-digestible polysaccharides, maintain intestinal health, regulate the immune system, provide an energy source.
*Lachnospiraceae*	Decompose a variety of plant fibers and polysaccharides, produce beneficial metabolites.
*Lactobacillus rhamnosus*	Enhance immune function, improve gut health.
*Lactobacillus rhamnosus* GG	Regulate immune response, enhance intestinal barrier function, and alleviate intestinal diseases.
*Firmicutes*	Extract energy and maintain intestinal health.
Bacteroidetes	Break down fiber and maintain immune regulation
*Akkermansia muciniphila*	Break down mucus, produce SCFAs, strengthen the intestinal barrier, and reduce inflammation.
*Escherichia coli*	Certain strains of *E. coli* can produce vitamin K and B group vitamins, participate in the nutritional intake, help to decompose nutrients
*Parvimonas micra*	Considered one of the main causative agents of periodontitis, and infections at other sites.
Adherent-invasive *Escherichia coli* (AIEC)	Adhesion and invasion of intestinal epithelial cells, leading to impairment of intestinal barrier function and triggering inflammatory response.

## Data Availability

Not applicable.

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
