# Peer review of "Extracellular Vesicles: A Crucial Player in the Intestinal Microenvironment and Beyond"

_ijms, 2024, doi:10.3390/ijms25063478_

Round 1

Reviewer 1 Report

Comments and Suggestions for Authors

The article is very interesting before, it could be considered for further publication I have some major quires which authors need to incorporate in the revised version of the manuscript for consideration of publication in the journal.

Title The title is Ok.

Abstract The last part needs to be rephrased. Authors wrote that “Lastly, we will discuss”, It is better to write “We discussed……”. The sentence starting with “Hopefully” needs to be re-writtin so as to bring clarity of the thoughts. Also, highlight essentialities and future perspectives of the study.

In the introductory section,

The authors need to correct “biological drug carriers for targeted delivery” and “biological agents for therapeutic objectives” The other problem is that no paper is cited for this. The authors have cited only one paper at the end of the para. The authors can consider the following papers:

Jan AT, Rahman S, Sarko DK, Redwan EM (2023). Exploring the role of exosomes in disease progression and therapeutics in neurodegeneration. Frontiers in Aging Neuroscience.15:1177063

Jan AT, Rahman S, Badierah R, Lee EJ, Mattar E, Redwan EM, Choi I. (2021) Expedition into exosome biology: A perspective of progress from discovery to therapeutic development. Cancers. 13(5):1157

The intro serves as a background to the study. The authors have cited only 3 papers to this. The unwarranted statements need to be supplemented by corresponding references.

In the Basics of Extracellular Vesicles section

The section needs a figure or a table depicting different components of the EVS.

The title EVs participate in intestinal regulation need to be changed to EVs in health and disease. The authors can the progressively move to EVs in intestinal regulation.

In Microbiota involved in and beyond the intestines section

The authors are advised to add a table summarizing different types of microbiota and their effect within the system.

Microbiome-Gut-Brain axis

The section is important for the manuscript. It needs elaboration with respect to different diseases.

Conclusion and perspectives

It is advised to make is crisp and clear. It is better to add references to literature of any suitable sources where used for correlation. Please update your conclusion in light of recent reports from 2019-24. Do add a short future perspective of the study.

Comments on the Quality of English Language

Minor correction is needed

Reviewer 2 Report

Comments and Suggestions for Authors

The review of Jahan et al. provides information about the current knowledge on extracellular vesicles in the intestinal micro- environment. Moreover, they discussed the roles of microbial and cellular EVs in the microbiome-gut-brain axis.  However, the manuscript requires major revisions to present the data in a clearer way.

Line 59. EVs also contain metabolites.

Line 93-121. The isolation methodologies of EVs are not necessary in this review. There are many reviews that describe these methodologies.

Line 128-131. What does “central regulator” mean? What is the relationship between the immune system and its defense against external pathogens?

Line 131-132. How does the gut microbiota aid in the digestion of food in the gastrointestinal tract?

Line 135-135. Authors state “receives exogenous EVs from ingested foods”. Does that also imply the microbiota derive EVs.

Line 140-142. The following sentence is difficult to understand: “the intestinal epithelia carry immune molecules that facilitates antigen presentation in immune cells, a mechanism that links antigen identification in the intestinal lumen to the immune system”

Line 147-150. From what origin does MSC-EVs come from?

Line 175-176. The sentence “distinguishing potential pathogens from antigens, rendering them as indispensable components in the intestinal immune system” is not clear. Pathogens should also contain some antigens.

Line 185: also, metabolites.

Intestinal EVs can be derived from microbiota and food such as plant exosome-like nanoparticles (ELNs). If the cargo of these EVs is released into intestine cells, it can be very dangerous to recipient cells. These external EVs can contain a lot of antigens in the form of proteins. Therefore, the relationship between external EVs, intestinal barrier and intestinal immune system is not clear. It is really difficult to believe that EVs originating from different species (like plants) can be transmitted to a host. I think that the authors should make a clear explanation about these relationships.

Line 260-264. The inverse process that EVs derived from intestine can “exert multiple functions” on “the git microbiota” is not clear. What functions?

Line 385-387. It will be really good to provide some proof for the following statement “The uptake of microbial EVs by intestinal epithelial cells occurs via various mechanisms, including actin-dependent macropinocytosis, etc” Could the authors provide references or figures that show how protein from microbial EVs is transferred to intestinal epithelial cells.

It is recommended that the authors provide some references or examples regarding microbiome-gut-brain axis with small molecules and proteins (providing name of metabolite and proteins). Also, the author should explain how these external proteins are not toxic for host cells.

Overall, in this review, I cannot understand where the hypothesis starts and where the proof for some of the conclusions are located.

Round 2

Reviewer 2 Report

Comments and Suggestions for Authors

The authors have clearly responded and corrected to my comments appropriately.

Author Response

Thank you.